Characteristics, source apportionment and health risks of indoor and outdoor fine particle-bound polycyclic aromatic hydrocarbons in Jinan, North China

Gao Xiaomei 1
Wang Ziyi 2
Sun Xiaoyan 3
Gao Weidong stu_gaowd@ujn.edu.cn 1
Jiang Wei 1
Wang Xi 1
Zhang Fenfen 1
Wang Xinfeng 2
Yang Lingxiao 2
Zhou Yang 4 5
1 School of Water Conservancy and Environment, University of Jinan , Jinan , Shandong Province , China
2 Environment Research Institute, Shandong University , Qingdao , Shandong Province , China
3 Jinan Ecological and Environmental Monitoring Center , Jinan , Shandong Province , China
4 Frontier Science Center for Deep Ocean Multispheres and Earth System (FDOMES) and Physical Oceanography Laboratory, Ocean University of China , Qingdao , Shandong Province , China
5 College of Oceanic and Atmospheric Sciences, Ocean University of China , Qingdao , Shandong Province , China
Ge Xinlei
Electronic publication date: 2024 Dec 16
Publication date: 2024
Volume: 12
Electronic Location ID: e18553
Received 2024 Jun 19; Accepted 2024 Oct 29
Copyright: ©2024 Gao et al.
Copyright year: 2024
Copyright holder: Gao et al.
License: This is an open access article distributed under the terms of the Creative Commons Attribution License, which permits unrestricted use, distribution, reproduction and adaptation in any medium and for any purpose provided that it is properly attributed. For attribution, the original author(s), title, publication source (PeerJ) and either DOI or URL of the article must be cited.
License URL: https://creativecommons.org/licenses/by/4.0/

Keywords: PAHs, PM2.5, Indoor, Outdoor, Sources, Health risk assessment, North China

Funding: National Natural Science Foundation of China 21607054 Natural Science Foundation General Project of Shandong ZR2022MD010 Doctoral Found Project XBS1429 University of Jinan XKY1326 This work was supported by the National Natural Science Foundation of China (No. 21607054), Natural Science Foundation General Project of Shandong (No. ZR2022MD010), and the Doctoral Found Project (No. XBS1429), and Scientific Research Fund (No. XKY1326) from University of Jinan. The funders had no role in study design, data collection and analysis, decision to publish, or preparation of the manuscript.

==============================
To investigate the pollution characteristics of polycyclic aromatic hydrocarbons (PAHs) indoors and outdoors and their influencing factors, PM2.5 samples were systematically collected from both environments in Jinan during the summer and autumn seasons. During the observation period, the concentration of ∑ 19PAHs was 18.57 ± 10.50 ng/m3 indoors and 23.79 ± 16.13 ng/m3 outdoors. Most PAHs exhibited indoor-to-outdoor (I/O) ratios less than 1, indicating that indoor PAHs were primarily derived from the infiltration of outdoor sources. Correlation analysis underscored the significant influence of temperature on both outdoor concentrations and I/O ratios of PAHs. By utilizing diagnostic ratios and principal component analysis (PCA), vehicle emissions were identified as the predominant source of outdoor PAHs. Our study found that the toxic equivalents of benzo[a]pyrene (TEQBaP) values exceeded the European Commission’s standard of 1 ng/m3, with indoor values at 2.78 ng/m3 and outdoor values at 3.57 ng/m3. Moreover, the total incremental lifetime cancer risk (ILCRTotal) associated with exposure to PM2.5-bound PAHs surpassed the acceptable level of 10E-6, indicating potential adverse health effects. These results underscore the urgent necessity for more stringent regulatory measures to reduce PAH emissions. Additionally, our findings provide valuable insights into how environmental factors shape the relationship between indoor and outdoor PAHs.

Introduction

Polycyclic aromatic hydrocarbons (PAHs) bound to fine particles (PM2.5) are highly concerning contaminants due to their resistance to degradation and detrimental impacts on ecosystems and living organisms (Kim et al., 2013; Sun et al., 2021). These PAHs are primarily released into the atmosphere through various processes such as incomplete combustion, pyrolysis of organic materials, and spills of liquid fuels (Abbas et al., 2018; Grmasha et al., 2023a; Zhang & Tao, 2009). Once these contaminants enter river basins via atmospheric deposition, they have the potential to accumulate within aquatic life, ultimately posing significant health risks to humans through the food chain (Grmasha et al., 2023b; Lee et al., 2021). Epidemiological studies have consistently established a link between both long- and short-term exposure to particulate PAHs and adverse health outcomes, including compromised pulmonary function, increased vulnerability to pulmonary diseases, and DNA damage (Barbosa et al., 2023; Li et al., 2019; Wang et al., 2019a). Given these concerns, the characterization of PAHs becomes paramount.

Extensive research has been conducted on PM2.5-bound PAHs, particularly in densely populated and industrialized regions of China (e.g., Hong et al., 2021; Liu et al., 2024; Shen et al., 2019; Sun et al., 2022; Ting et al., 2024; Xu et al., 2024; Zhang, Chen & Lv, 2016; Zhang et al., 2021a; Zhou et al., 2022). These studies have revealed significant seasonal variations, with higher concentrations observed during winter, which was influenced by emission sources and meteorological factors such as temperature and wind speed (Ma et al., 2020; Wang et al., 2019b; Zhang et al., 2019a; Zhu et al., 2015). Vehicle emissions, coal combustion, and biomass burning have been identified as the primary sources, and potential health risks associated with exposure to PM2.5-bound PAHs have also been established. However, despite the extensive research on outdoor PAHs, humans spend a considerable amount of time indoors, where poor indoor air quality has been estimated to contribute significantly to global disease risk (Ma & Harrad, 2015). A cross-sectional study by Singh et al. (2016) revealed a correlation between diminished lung function among workers and relatively high PAH concentrations in kitchen environments. Consequently, the contamination of indoor environments by PAHs has emerged as a critical issue that requires immediate attention.

Indoor PM2.5-bound PAHs are influenced by various sources, including outdoor environments (e.g., coal combustion, traffic emissions, and biomass burning) and endogenous sources (e.g., cooking, smoking, and candle burning) (Castro et al., 2011; Chen et al., 2017; Orecchio, 2011; Shen et al., 2022). Previous studies conducted in various settings, such as in the office and hotel of Jinan (Li et al., 2017; Zhu et al., 2015), in the office, residential apartment, students’ dormitory, and primary school classroom of Beijing (Chen et al., 2017; Han et al., 2016; Zhang et al., 2020), and students’ laboratory of Caofeidian (Zhang et al., 2021b), have observed a positive correlation between indoor and outdoor PAHs in PM2.5, with lower concentrations indoors compared to outdoors. These findings suggested that indoor PAHs primarily depended on the influx of outdoor particles when significant indoor combustion sources were absent. The transport of outdoor particles into indoor environments is significantly influenced by meteorological conditions (Isaacs et al., 2013; Wang et al., 2024). Although these studies have examined the relationship between indoor and outdoor PAHs under mechanical/natural ventilation conditions and human activities, the influence of meteorological factors on this relationship remains relatively under-researched.

As the capital of Shandong Province located in the North China Plain, Jinan has long struggled with considerable PM2.5 pollution. In response to this challenge, the government launched a series of clean air actions from 2013 to 2017, resulting in a notable reduction in PM2.5 concentrations across the city (Gao et al., 2020; Li et al., 2022). Previous studies on PM2.5-bound PAHs in Jinan were primarily conducted on the pre-2017 period, such as 2010, 2015 and 2016 (Li et al., 2017; Zhang et al., 2018; Zhu et al., 2015). However, post-2017 studies on PAHs in Jinan have been relatively scarce (Cao et al., 2021), particularly regarding indoor PAHs. In this study, we collected PM2.5 samples from both indoor and outdoor environments in Jinan during 2019. Our primary objectives were to assess the characteristics, identify potential sources and evaluate health risks of PAHs. Additionally, we attempted to investigate the influence of meteorological factors on the relationship between indoor and outdoor PAHs in a vacant office with the door and windows closed in order to eliminate any human-related activities.

Materials & Methods

Sample collection

Simultaneous measurements were conducted in an office room and the outdoor atmosphere of an urban area in Jinan, China (36°69′N, 117°06′E). The outdoor sampling site was situated on the rooftop of a six-story office building on the west campus of University of Jinan, approximately 20 m above ground level. For indoor sampling, a vacant office room on the second floor of the same building, equipped with a door and two windows, was selected. During the sampling period, the windows were kept closed, ensuring no natural or mechanical ventilation occurred. The door remained closed throughout the process, except when students entered to collect the samples. Additional detailed information about the sampling site can be found in our previous study (Gao et al., 2020). Figure S1 shows the location of the sampling point and the picture of the vacant office where the sampling was conducted.

PM2.5 samples were collected during two seasons in 2019: a summer campaign from June 21st to July 5th, and an autumn campaign from October 26th to November 13th. These samples were manually collected using quartz filters with a 90 mm diameter and a 2 µm pore size (Pall, Port Washington, NY, USA) and an intelligent sampler (Model TH-150A; Wuhan Tianhong Corporation, China) operating at a flow rate of 100 L/min. Typically, PM2.5 samples were collected for 24 h from 8:00 a.m. to 7:30 a.m. the following day. A total of 31 sample sets (containing 13 sets in summer and 18 sets in autumn) and 4 sets of blank samples were obtained for both indoor and outdoor environments. To eliminate any background organic matter, the quartz filters were pretreated at a temperature of 450 °C for 6 h before sample collection. After sampling, all filters were stored in plastic Petri dishes and refrigerated at −4 °C.

Chemical analysis

Three quartz filters, each with a diameter of 25 mm, were excised and placed within a 10 mL stainless steel extraction column. A solvent mixture consisting of n-hexane and dichloromethane (1:1-v/v) was introduced, and the extraction process was repeated twice, each for 15 min, at a temperature of 100 °C and a pressure of 11 MPa. The extracted solution was then concentrated to 1–2 mL using a rotary evaporator and further purified through a solid-phase extraction (SPE) tube (Supelclean ENVI-Florisil SPE, SUPELCO, USA). Following purification, the eluent was evaporated to 0.5–1 mL by rotation, transferred to a brown vial, and adjusted to a final volume of one mL. Prior to analysis, 100 ng of deuterated PAHs, including naphthalene-d8 and anthracene-d10, were added as internal standards for quantitative assessment.

Detection of PAHs was performed using gas chromatography coupled with a single quadrupole mass spectrometry (TRACE1300-IQS7000, Thermo-Fisher, Waltham, MA, USA). The GC oven temperature program was as follows: an initial hold at 60 °C for 1 min, a ramped to 160 °C at a rate of 20 °C/min for 1 min, a further increased to 280 °C at 6 °C/min for 5 min, and a final elevation to 300 °C at 20 °C/min for 10 min. The ion source was operated in electron impact (EI) mode at 70 eV. The resulting chromatogram, presented in Fig. S2, demonstrates the quantification of nineteen PAHs: naphthalene (NAP), biphenyl (BYL), acenaphthylene (ACY), acenaphthene (ACE), fluorene (FLU), phenanthrene (PHE), anthracene (ANT), fluoranthene (FLT), pyrene (PYR), benzo[a]anthracene (BaA), chrysene (CHR), benzo[b]fluoranthene (BbF), benzo[k]fluoranthene (BkF), benzo[a]pyrene (BaP), indeno[1,2,3-cd]pyrene (IcdP), dibenzo[a,h]anthracene (DahA), benzo[g,h,i]perylene (BghiP), benzo[e]pyrene (BeP), and coronene (Cor).

Risk assessment

The risk assessment of potential health impacts from exposure to PM2.5-bound PAHs was conducted in terms of carcinogenic equivalent toxicity (TEQBaP), noncarcinogenic risk (non-CR), and incremental lifetime cancer risk (ILCR). These assessments were considered variations in physiological characteristics and living habits among different age groups (children aged 1–11, adolescents aged 12–17, and adults aged 18–70) and genders (Li et al., 2022; Ma et al., 2020). TEQBaP was determined by summing the products of the PAH concentrations (Ci) and their respective toxicity equivalent factors (TEFi) relative to BaP, as outlined in Eq. (1). (1) ∑TEQBaP= ∑Ci×TEFi.

Non-CR was evaluated through the Hazard Index (HI) for inhalation exposure, as defined in Eq. (2). (2) HITotal= ∑n=19∑in+outEF×Ci×IRInhalation×EDBW×AT×RfDi×106

The ILCR, resulting from inhalation, ingestion, and dermal contact, was calculated using Eqs. (3)–(6): (3) ILCRInhalation=CSFInhalation×BW703×IRInhalation×EDBW×AT×PEF×∑in+outEF×∑TEQBaP

(4) ILCRIngestion=CSFIngestion×BW703×IRIngestion×EDBW×AT×106×∑in+outEF×∑TEQBaP

(5) ILCRDermal=CSFDermal×BW703×SA×AF×ABS×EDBW×AT×106×∑in+outEF×∑TEQBaP

(6) ILCRTotal=ILCRInhalation+ILCRIngestion+ILCRDermal.

In these equations, Ci represents the concentration of individual PAHs (ng/m3) in PM2.5; IRInhalation represents the inhalation rate (m3/d); ED represents the exposure duration (year); EF denotes the exposure frequency (d/year); BW represents the body weight (kg) of the individual; AT stands for the average lifespan (year); RfDi represents the oral cancer slope factor of each PAHs compound (mg/(kg day)); PEF is the soil dust produce factor (m3/kg); CSFInhalation, CSFIngestion, and CSFDermal are the carcinogenic slope factors for inhalation, ingestion and dermal exposure, respectively; IRIngestion is the soil intake rate (mg/d); SA represents the dermal surface exposure (cm2/d); AF denotes the dermal adherence factor (mg/cm2); ABS is the dermal adsorption fraction. In this assessment, it was assumed that individuals spend 80% of their time indoors, with an exposure frequency is 365 d/year. The specific parameters used in these equations are detailed in Tables S1 and S2 (Bai et al., 2023; Li et al., 2022; Ma et al., 2020).

Quality assurance/control

In this study, quality assurance and control measures were implemented to ensure the accuracy and reliability of the analytical results (Grmasha et al., 2024a). The experimental setup was meticulously designed to minimize contamination, utilizing analytical-grade n-hexane and dichloromethane (DCM) solvents specifically selected for the pretreatment process. Glassware was thoroughly cleaned with chromic acid, baked at 100 °C, and oven-dried to eliminate potential contaminants. To ensure instrument stability and consistent performance, a PAH calibration solution was analyzed after every ten samples. Data quantification was performed using an internal calibration method with five-point calibration curves, achieving high linearity with R2 values exceeding 0.99 for each PAH. Surrogate recoveries were evaluated by spiking one-third of the samples with 100 ng of acenaphthene-d10 and chrysene-d12 before extraction, resulting in recoveries of 103% ± 8% and 72% ± 7%, respectively. The limits of detection (LOD) were determined based on the analyte concentration, using a three-fold signal-to-noise ratio (Grmasha et al., 2024b). The LODs for the 19 PAHs ranged from 0.1 to 10 µg/L. To assess extraction efficiency, spiked recovery experiments were conducted using blank filters and polyurethane foam (PUF), yielding average PAH recoveries of 73%–98% and 75%–105%, respectively. Additionally, to validate the reproducibility of the analyses, replicate measurements were performed every ten samples, demonstrating a variation of 0.1%–20.7% for the 19 PAHs.

Results & Discussion

Concentration and chemical composition

The mass concentrations of PM2.5 and 19 PAHs in both indoor and outdoor environments are presented in Table 1 and Fig. 1. During the sampling periods, the concentrations of the sum of 19 PAHs (∑19PAHs) were measured at 18.57 ± 10.50 ng/m3 indoors and 23.79 ± 16.13 ng/m3 outdoors. These concentrations were 2 to 3 times higher during the autumn season compared to the summer season. As detailed in Table S3, the outdoor concentrations of PAHs in our study were higher than those reported in various cities such as Beijing (Li et al., 2022; Chen et al., 2017), Dongguan (Chen et al., 2022), Shanghai (Hong et al., 2021; Yang et al., 2021), Nanjing (Hong et al., 2021), Taipei (Ting et al., 2024), as well as Kanazawa, Japan and Auckland, New Zealand (Kalisa et al., 2019), and Rome, Italy (Romagnoli et al., 2014). However, they were lower than the PAH levels observed in Xi’an (Wang et al., 2019b), Harbin (Ma et al., 2020), and Tehran, Iran (Ali-Taleshi et al., 2021), and comparable to those in Wuhan (Zhang et al., 2019b) and Caofeidian (suburban) (Zhang et al., 2021b). Indoor PAH pollution was also modest when compared to indoor PAH levels reported in other studies (see Table S4).

Table 1 Average concentrations of PM2.5 (µg/m3) and 19 PAHs (ng/m3) monitored in indoor and outdoor air samples during the summer and autumn seasons.

	Compound	Overall (n = 31)	Summer (n = 13)	Autumn (n = 18)	
		Indoor	Outdoor	Indoor	Outdoor	Indoor	Outdoor	
	PM2.5	45.18 ± 16.62	65.95 ± 31.48	38.14 ± 9.18	40.29 ± 11.29	50.27 ± 19.04	84.48 ± 28.14	
2ring	NAP	0.70 ± 0.20	0.76 ± 0.33	0.63 ± 0.16	0.61 ± 0.20	0.76 ± 0.22	0.88 ± 0.36	
2ring	BYL	0.35 ± 0.11	0.35 ± 0.12	0.28 ± 0.06	0.29 ± 0.07	0.39 ± 0.12	0.39 ± 0.13	
3ring	ACY	0.20 ± 0.05	0.22 ± 0.07	0.17 ± 0.03	0.17 ± 0.03	0.22 ± 0.04	0.25 ± 0.07	
3ring	ACE	0.19 ± 0.06	0.21 ± 0.07	0.17 ± 0.03	0.18 ± 0.04	0.21 ± 0.06	0.23 ± 0.08	
3ring	FLU	0.30 ± 0.07	0.34 ± 0.12	0.27 ± 0.05	0.28 ± 0.08	0.33 ± 0.07	0.39 ± 0.12	
3ring	PHE	0.84 ± 0.36	0.99 ± 0.57	0.50 ± 0.13	0.50 ± 0.20	1.09 ± 0.25	1.34 ± 0.47	
3ring	ANT	0.22 ± 0.05	0.23 ± 0.08	0.19 ± 0.04	0.18 ± 0.03	0.24 ± 0.06	0.27 ± 0.08	
4ring	FLT	1.57 ± 0.86	1.91 ± 1.22	0.76 ± 0.18	0.73 ± 0.29	2.15 ± 0.65	2.76 ± 0.87	
4ring	PYR	1.21 ± 0.67	1.53 ± 1.00	0.59 ± 0.14	0.59 ± 0.22	1.66 ± 0.51	2.22 ± 0.74	
4ring	BaA	0.69 ± 0.31	0.89 ± 0.56	0.43 ± 0.08	0.43 ± 0.11	0.89 ± 0.27	1.23 ± 0.51	
4ring	CHR	1.23 ± 0.67	1.76 ± 1.19	0.68 ± 0.17	0.78 ± 0.29	1.64 ± 0.59	2.46 ± 1.10	
5ring	BkF	2.79 ± 1.83	3.77 ± 3.10	1.34 ± 0.39	1.51 ± 0.68	3.84 ± 1.75	5.40 ± 3.16	
5ring	BbF	0.85 ± 0.46	1.10 ± 0.78	0.49 ± 0.11	0.56 ± 0.18	1.12 ± 0.43	1.50 ± 0.81	
5ring	BeP	1.45 ± 0.92	1.96 ± 1.51	0.72 ± 0.21	0.86 ± 0.37	1.97 ± 0.87	2.75 ± 1.53	
5ring	BaP	1.64 ± 1.13	2.14 ± 1.77	0.74 ± 0.19	0.87 ± 0.31	2.29 ± 1.07	3.05 ± 1.83	
5ring	DahA	0.51 ± 0.18	0.59 ± 0.28	0.36 ± 0.04	0.42 ± 0.09	0.57 ± 0.18	0.69 ± 0.30	
6ring	IcdP	1.62 ± 1.26	2.15 ± 1.85	0.69 ± 0.20	0.87 ± 0.34	2.30 ± 1.26	3.07 ± 1.96	
6ring	BghiP	1.69 ± 1.27	2.21 ± 1.84	0.73 ± 0.21	0.95 ± 0.39	2.39 ± 1.27	3.13 ± 1.93	
7ring	Cor	0.60 ± 0.37	0.76 ± 0.56	0.35 ± 0.06	0.42 ± 0.13	0.77 ± 0.39	0.99 ± 0.62	
	∑19PAHs	18.57 ± 10.50	23.79 ± 16.13	9.92 ± 2.27	11.05 ± 3.86	24.82 ± 9.61	32.99 ± 15.33	

In comparison to previous studies conducted in Jinan, the concentrations in this study were lower (See Tables S3 and S4). For example, Zhu et al. (2015) documented outdoor concentrations of 30.78 ng/m3 and 140.34 ng/m3, and indoor concentrations of 29.20 ng/m3 and 111.07 ng/m3 during the summer and autumn of 2010, respectively. Additionally, a study conducted in 2015 reported an outdoor concentration of 19.45 ng/m3 during summer (Zhang et al., 2018), while another study in 2016 during winter reported an outdoor concentration of 105.30 ng/m3 and indoor concentrations ranging from 63.26 ng/m3 to 115.63 ng/m3 (Li et al., 2017). Furthermore, the outdoor concentration in this study was comparable to that reported in Jinan in 2016 (Zhang et al., 2019a). The observed decrease in concentrations may suggest the effectiveness of anti-pollution strategies implemented in China.

Figure 1 The concentrations of PAHs ng/m3.

I and O represent indoor and outdoor, respectively.

The composition profiles of PAHs in indoor and outdoor environments exhibited consistent patterns, as illustrated in Fig. 2. Among the PAHs analyzed, BkF consistently emerged as the most abundant, followed by BaP, BghiP, IcdP, FLT, and BeP. These six species collectively accounted for approximately 50% to 60% of the total PAH concentration. The distribution of PAHs based on the number of aromatic rings, depicted in Fig. 2B, indicated that 5-ring PAHs predominated and contributed around 35% to 40% to the total PAHs, followed by 4-ring and 6-ring PAHs. This observation suggested that the sampling site was influenced by vehicle emissions during the sampling period (Chen et al., 2017; Tang et al., 2017).

Figure 2 The average proportions of components (A) and different rings (B) of PAHs.

In this study, Spearman correlation analysis was used to investigate the relationships between meteorological factors and outdoor PAH levels, as shown in Table 2. The results showed a significant negative correlation with temperature (r = −0.70, p < 0.05), while there was no significant correlation with wind speed (r = −0.16, p > 0.05), suggesting that temperature had a stronger influence on PAH concentrations compared to wind speed. During the study period, a negative correlation was observed between temperature and all PAHs, with correlation coefficient (r) ranging from −0.36 to −0.74 (p < 0.05). The summer temperature averaged 33 °C ±2 °C, which was 10 °C higher than the autumn temperature of 21 °C ±1 °C. The elevated summer temperature facilitated the transition of PAHs from the particulate phase to the gas phase and promoted their secondary transformation, resulting in decreased PAH concentrations in summer compared to autumn. However, it is crucial to note that this process was likely more complex than solely a temperature difference. LMW PAHs (2–ring and 3–ring) largely exist in the gas phase during high summer temperatures due to their high vapor pressure (Araki et al., 2009; Zhang et al., 2020; Zhu et al., 2022), which would typically lead to lower proportions of LMW PAHs in summer compared to autumn. Contrary to expectations, the observed proportions of 2–ring and 3–ring PAHs in the total PAHs were 8.68% and 12.69% in summer, respectively, and decreased to 4.12% and 8.15% in autumn. Therefore, factors other than temperature should also be considered. Zhu et al. (2015) proposed that emissions from biomass and agricultural waste burning significantly contributed to increased PAH concentrations during autumn. Several studies have suggested that emission sources played a predominant role in driving changes in PAH concentrations, as evidenced by a positive correlation between PAH and PM2.5 concentrations (Hu et al., 2012; Zhu et al., 2015). As shown in Table 2, a strong positive correlation (ranging from 0.50 to 0.81, p < 0.01) was observed between PM2.5 and PAHs. Moreover, the relationship between PAHs with PM2.5 was found to be stronger than that with temperature, indicating that emission sources likely have a greater influence on the variability of PAH concentrations compared to gas-particle partitioning processes. These findings highlighted the importance of implementing emission control measures to mitigate levels in the environment.

Table 2 Spearman correlation coefficient (two-tailed) of PAHs with PM2.5, temperature, and wind speed for outdoors during the campaigns (n = 31).

	PM2.5	Temperature	Wind speed	
NAP	0.50**	−0.36*	−0.01	
BYL	0.51**	−0.42*	−0.17	
ACY	0.66**	−0.60**	−0.17	
ACE	0.57**	−0.42*	−0.03	
FLU	0.57**	−0.46*	−0.14	
PHE	0.77**	−0.72**	−0.13	
ANT	0.67**	−0.62**	−0.08	
FLT	0.81**	−0.67**	−0.03	
PYR	0.80**	−0.72**	−0.12	
BaA	0.77**	−0.73**	−0.20	
CHR	0.77**	−0.74**	−0.19	
BkF	0.78**	−0.65**	−0.13	
BbF	0.77**	−0.63**	−0.09	
BeP	0.78**	−0.68**	−0.15	
BaP	0.75**	−0.68**	−0.17	
DahA	0.63**	−0.42*	−0.07	
IcdP	0.78**	−0.63**	−0.12	
BghiP	0.77**	−0.67**	−0.15	
Cor	0.70**	−0.52**	−0.13	
LMW	0.69**	−0.59**	−0.08	
MMW	0.79**	−0.72**	−0.14	
HMW	0.77**	−0.66**	−0.14	
∑19PAHs	0.78**	−0.70**	−0.16	
Notes.

Level of significance: *: p <  0.05; **: p <  0.01.

Relationship between indoor and outdoor PAH concentrations

In the office room, PAHs exhibited seasonal variations and chemical composition profiles that closely mirrored those observed at the outdoor site. To investigate the relationship between indoor and outdoor PAH concentrations, we conducted correlation analyses and calculated indoor-to-outdoor (I/O) ratios, with the results summarized in Table S5 and Fig. 3. The mean I/O ratios for all PAHs, except for BYL, were found to be less than 1.00. Furthermore, a significant positive correlation was observed between indoor and outdoor PAH levels (r = 0.55–0.93, p < 0.01), as detailed in Table S5. No significant differences were detected between indoor and outdoor PAH concentrations, with mean concentration deviations being less than 25% and p values > 0.05 for most PAHs. These findings indicated a substantial influence of outdoor sources on indoor PAH concentrations during the study period. The I/O ratios of PAHs decreased in the order of LMW, MMW, and HMW PAHs. Conversely, the correlation coefficients between indoor and outdoor PAHs increased with molecular weight, yielding values of r = 0.78, 0.85, and 0.89 for LMW, MMW, and HMW PAHs, respectively. These results suggested that LMW PAHs were relatively more influenced by indoor sources compared to MMW and HMW PAHs, although indoor PAHs primarily originated from the outdoor atmosphere (Chen et al., 2017). In comparison with the previous study (Zhu et al., 2015), in this study, the I/O ratios for LMW and MMW PAHs were lower, whereas those for HMW PAHs were higher. This finding suggested that indoor sources had a less significant impact on LMW and MMW PAHs in our study, while more HMW PAHs infiltrated indoors from outdoors.

Figure 3 I/O ratios for PAHs.

(Boxes range 25% and 75%; Hollow square mean; Whiskers 1.5 times the inter-quartile range away).

We also investigated the influence of meteorological factors on the I/O ratios. A positive correlation was observed between I/O ratios and temperature (r = 0.31, p > 0.05) as well as wind speed (r = 0.18, p > 0.05) (Table S6), although these correlations were not statistically significant. This implied that temperature and wind speed may facilitate the infiltration of PAHs from the outdoor environment into indoor spaces. It is noteworthy that measurements were conducted with doors and windows closed, indicating that infiltration through cracks and leaks was the primary pathway for outdoor particles entering indoors (Hou et al., 2019). Further analysis was conducted by segregating the effects of temperature and wind speed (Fig. 4). As shown in Fig. 4, when temperature exceeded 20 °C, I/O ratios increased with rising temperatures, and a significant correlation between I/O ratios and temperature was also observed (r = 0.59, p < 0.01). Due to relatively low wind speeds, the data were categorized into two groups based on a threshold of 1.5 m/s. Higher wind speeds correlated with higher I/O values (Fig. 4). These results showed that temperature and wind speed influenced the import of outdoor-originated air into indoor environments (Wang et al., 2024), and further investigation will be needed to elucidate their effects more clearly.

Figure 4 The relationship between I/O ratios with temperatures (A), and wind speed (B).

(Boxes range 25% and 75%; Hollow square mean; Whiskers 1.5 times the interquartile range away).

Source analysis

Considering the substantial impact of outdoor sources on indoor PAHs, we employed diagnostic ratios and principal component analysis (PCA) to identify the sources of outdoor PAHs in this study.

PAHs source identification by diagnostic ratios

Diagnostic ratios have been widely used in various studies to qualitatively identify the potential sources of PAHs (e.g., Grmasha et al., 2022; Ting et al., 2024; Tobiszewski & Namiesnik, 2012; Yunker et al., 2002). However, it is crucial to interpret diagnostic ratios with caution, as their values can be influenced by phase transfers and environmental degradation in the atmosphere (Tobiszewski & Namiesnik, 2012). To mitigate potential misinterpretations, Tobiszewski & Namiesnik (2012) recommended using multiple diagnostic ratios for validation purposes. Therefore, we adopted six diagnostic ratios and compared them to the reference values to determine the possible sources of PAHs, and the results are shown in Table 3.

Table 3 Diagnostic ratios of PAHs used as source indicator.

Diagnostic ratio	Mean ±SD	Min	Max	Source assignment	Reference	
ANT/(ANT+PHE)	0.22 ± 0.07	0.10	0.46	<0.1 Non-burned petrogenic source	Ting et al. (2024)	
>0.1 Combustion source	Tobiszewski & Namiesnik (2012)	
BaP/BghiP	0.99 ± 0.24	0.74	1.97	>0.6 Vehicle emissions	Ting et al. (2024)	
<0.6 Non-traffic emissions	Grmasha et al. (2022)	
BaA/(BaA+CHR)	0.35 ± 0.04	0.27	0.47	<0.2 Petroleum combustion		
0.2-0.35 Coal combustion	Hong et al. (2021)	
>0.35 Vehicle emissions		
IcdP/(IcdP+BghiP)	0.49 ± 0.03	0.41	0.61	<0.2 Petrogenic source		
0.2-0.5 Petroleum combustion	Zhang et al. (2021b)	
>0.5 Biomass and coal burning		
FLU/(FLU+PYR)	0.23 ± 0.11	0.12	0.55	<0.5 Diesel vehicles	Tobiszewski & Namiesnik (2012)	
>0.5 Gasoline vehicles	
PYR/BaP	0.78 ± 0.29	0.43	1.86	∼1 Gasoline emission	Grmasha et al. (2022)	
∼10 Diesel emission	

The ratio of ANT/(ANT + PHE) <0.1 indicates a non-burned petrogenic source, while a ratio >0.1 indicates a combustion source (Ting et al., 2024; Tobiszewski & Namiesnik, 2012). The ANT/(ANT + PHE) ratios ranged from 0.10 to 0.46, implying that PM2.5-bound PAHs were mainly derived from various combustion processes.

The ratios of BaA/(BaA + CHR) and IcdP/(IcdP + BghiP) can identify petroleum sources (FLA/(FLA + PYR) and IcdP/(IcdP + BghiP) < 0.20), combustion of petroleum (BaA/(BaA + CHR) = 0.20–0.35 or IcdP/(IcdP + BghiP) = 0.20–0.50), and biomass and coal combustion (FLA/(FLA + PYR) >0.35 or IcdP/(IcdP + BghiP) >0.50) (Yunker et al., 2002; Zhang et al., 2021b). The IcdP/(BghiP + IcdP) and BaA/(BaA + CHR) ratios were 0.35 ±  0.04 (ranging from 0.27 to 0.47) and 0.49 ± 0.03 (ranging from 0.41 to 0.61), respectively, suggesting mixed sources of petroleum and biomass/coal combustion.

The ratio of BaP/BghiP can distinguish traffic emissions (BaP/BghiP > 0.6) from non-traffic emissions (BaP/BghiP < 0.6) (Grmasha et al., 2022; Ting et al., 2024; Tobiszewski & Namiesnik, 2012). The BaP/BghiP ratios were more than 0.60, with an average of 0.99, indicating traffic emissions. Additionally, the ratios of FLU/(FLU+PYR) and PYR/BaP can identify sources of diesel and gasoline. As shown in Table 3, the FLU/(FLU+PYR) ratios were lower than 0.50 with an average of 0.23, and the PYR/BaP ratios were around 1, indicating that PAHs originate from gasoline combustion. Previous research has linked the presence of 5–ring and 6–ring PAHs species (such as BkF, BaP, IcdP, BghiP) to gasoline engine emissions, while LMW species are associated with diesel vehicle exhaust (Wu et al., 2014). Consistent with these findings, the major PAH components identified in this study were BkF, BaP, IcdP, and BghiP, providing further evidence that the sampling site was primarily influenced by gasoline-related emissions.

PAHs source apportionment by PCA

PCA is a robust data reduction technique widely employed to identify the sources of PAHs (Chen et al., 2017; Grmasha et al., 2024b; Hu et al., 2020; Zhu et al., 2014). PCA analysis uncovered three key factors that jointly explain a remarkable 95.66% of the total variance, and the results are shown in Table 4 and Fig. S3.

Table 4 The results from PCA analysis.

	Factor 1	Factor 2	Factor 3	
NAP	0.29	0.84	0.24	
BYL	0.17	0.88	0.39	
ACY	0.49	0.76	0.30	
ACE	0.13	0.95	0.16	
FLU	0.15	0.84	0.44	
PHE	0.25	0.40	0.87	
ANT	0.40	0.79	0.45	
FLT	0.41	0.37	0.81	
PYR	0.40	0.43	0.80	
BaA	0.70	0.35	0.60	
CHR	0.63	0.30	0.70	
BkF	0.83	0.26	0.48	
BbF	0.83	0.29	0.47	
BeP	0.81	0.28	0.50	
BaP	0.77	0.33	0.43	
IcdP	0.85	0.25	0.46	
DahA	0.83	0.43	0.25	
BghiP	0.85	0.24	0.45	
Cor	0.89	0.28	0.31	
Variance	78.93%	12.17%	4.56%	

Factor 1 exhibited a strong correlation with PAHs containing 5, 6, and 7 rings, including BkF, BbF, BeP, BaP, IcdP, DahA, BghiP, and Cor, all of which exhibited factor loadings exceeding 0.70. It is well-established that HMW PAHs are principally associated with vehicular sources (Wu et al., 2014). Previous studies have specifically highlighted the significant contribution of gasoline-powered vehicles to elevated levels of BbF, BaP, IcdP, DahA, and BghiP (Khan et al., 2015; Zhao et al., 2020). Additionally, Cor has been linked to gasoline vehicle emissions in studies by Guo et al. (2003) and Zhang et al. (2019b). Diesel emissions have also been identified as a substantial source of BkF, BaP, IcdP, and BghiP (Wu et al., 2014). Considering the sampling site, factor 1 was attributed to vehicular sources, explaining a significant portion (78.9%) of the observed variance.

Factor 2 was primarily associated with volatilization processes, particularly the evaporation of crude oil and petroleum products. LMW PAHs such as NAP, BYL, ACY, ACE, FLU, and ANT are known to be released through these processes (Abbas et al., 2018; Hong et al., 2021; Yunker et al., 2002). Elevated temperatures during the summer and autumn seasons facilitate the evaporation of gasoline, and a nearby gas station located about 500 m away from the sampling site could act as a source of oil evaporation and dispersion. Therefore, factor 2 likely represented PAHs released via volatilization processes, accounting for 12.2% of the observed variance.

Factor 3 was characterized by high loadings of FLT, PYR, PHE, and CHR which are associated with emissions from coal-fired power stations and industrial facilities that utilize coal as a fuel (Hong et al., 2021; Soledad Callen, Iturmendi & Manuel, 2014). FLT, PYR, and CHR are indicative of coal combustion, while FLT and PYR also serve as indicators for biomass combustion (Atzei et al., 2019; Jenkins et al., 1996). Thus, factor 3 represented a source profile related to coal and biomass combustion processes, explaining 4.56% of the observed variance.

Based on diagnostic ratios and PCA analysis, it was concluded that vehicle emissions were the primary source of PAHs during the summer and autumn seasons. These findings underscore the significant impact of vehicular emissions on PAH levels in the urban environment of Jinan.

Health risk assessment

The TEQBaP, HI, and ILCR values during the sampling periods are detailed in Tables S7–S9. All these values were higher in autumn compared to summer, aligning with the seasonal variation in PAH concentrations. The average TEQBaP values were 2.78 ng/m3 indoors and 3.57 ng/m3 outdoors, significantly surpassing the European Commission’s standard of 1 ng/m3 (European Union, EU). In comparison to analogous studies conducted in other cities, our TEQBaP values were lower than those reported in Beijing (30 ng/m3) (Wang et al., 2016), Urumqi (19 ng/m3) (Maimaiti et al., 2018), Baoji (8.6 ng/m3) (Zhang & Kong, 2020), and Liaocheng (6.2 ng/m3) (Liu et al., 2019), as well as previous studies in Jinan (Zhang et al., 2018; Zhu et al., 2015). However, they were higher than the levels observed in Tuoji Island (0.33 ng/m3), Mt. Tai (0.43 ng/m3) (Zhang et al., 2018), the Yellow River Delta National Nature Reserve (1.84 ng/m3) (Zhu et al., 2014), and Taipei (0.096 ng/m3) (Ting et al., 2024). The primary contributors to the TEQBaP values were HMW PAHs, particularly BaP, which accounted for approximately 60%, indicating a considerable health risk. BaP, a well-known human carcinogen, exhibited concentrations of 1.64 ± 1.13 ng/m3 indoors and 2.14 ± 1.77 ng/m3 outdoors. Although this value fell below China’s Ambient Air Quality Standards limit of 2.5 ng/m3 (GB3095–2012), it exceeded the World Health Organization’s limit of 1 ng/m3. This exceedance indicated the potential health risks posed by PAHs in Jinan.

To assess non-cancer risks, the Hazard Index (HI) was calculated for nine PAHs. The total HI (HITotal) values ranged from 2.57E-4 to 6.90E-4, with BaP being the primary contributor, accounting for 94.28% of the HITotal (Fig. 5). This finding underscored BaP’s significant non-cancer risk potential compared to the other studied PAHs. When stratified by age, adulthood (both males and females) exhibited the highest HI values, which were 1.76 to 2.02 times higher than those for adolescence and 2.35 to 2.45 times higher than childhood. A gender-based analysis revealed minor differences in HI values, with female adults having slightly lower values than male adults. However, these differences were not statistically significant (p = 0.92∼0.96).

Figure 5 HI values of PAHs.

1, 2, 3, 4, 5, and 6 represent childhood male, childhood female, adolescence male, adolescence female, adulthood male, and adulthood female, respectively.

Regarding cancer risks, the total incremental lifetime cancer risk (ILCRTotal) values resulting from exposure to PM2.5-bound PAHs exceeded the acceptable level of 10E-6 (see Fig. 6), suggesting a potential cancer risk (Li et al., 2022; Zhang et al., 2021b). Female adults were found to have the highest potential cancer risk, followed by female adolescents, male adults, male adolescents, female children, and male children. This pattern aligns with HI results, suggesting that females generally face a higher cancer risk across all exposure pathways. Interestingly, adolescents’ probabilistic cancer risk was comparable to that of adults, despite their shorter exposure durations. The primary exposure routes leading to elevated cancer risk were ingestion and dermal contact, with values several orders of magnitude higher than inhalation.

Figure 6 ILCR values of PAHs.

1, 2, 3, 4, 5, and 6 represent childhood male, childhood female, adolescence male, adolescence female, adulthood male, and adulthood female, respectively.

It is imperative to recognize the underestimations in the reported risk values, as emphasized by our study. Primarily focusing on the summer and autumn seasons, our investigation did not consider the typically higher concentrations of PAHs observed during winter in North China (Chen et al., 2017; Hong et al., 2021; Li et al., 2022; Zhu et al., 2014). This seasonal variation suggests that the actual risk values during these periods could exceed those presented in our findings. Moreover, our research was limited to assessing exposure within an office setting, excluding other indoor environments where exposure occurs. This limitation restricts the generalizability of our findings. Additionally, our analysis only considered a subset of PAH species in terms of toxicity. Importantly, data regarding the toxicity of certain carcinogenic species, including NPAHs and OPAHs, were not included in our study (Li et al., 2022). Incorporating the toxicity of these additional PAH species would enhance our understanding of the potential cancer risks associated with exposure to PM2.5-bound PAHs, providing a more comprehensive evaluation of the hazards posed by these pollutants.

The health risk assessment results indicated that both carcinogenic and non-carcinogenic risks were not negligible, despite the aforementioned underestimations due to the restricted available data. These findings underscore the necessity for future studies to account for seasonal variations to achieve a more robust and comprehensive health risk assessment.

Conclusions

The study conducted in Jinan, China, provided profound understanding of the characteristics, sources, and health risks of PM2.5-bound PAHs in both indoor and outdoor environments in summer and autumn. The concentration of ∑19 PAHs was determined to be 18.57 ± 10.50 ng/m3 indoors and 23.79 ± 16.13 ng/m3 outdoors, and it was 2.50 and 2.99 times higher, respectively, in autumn compared to summer. The PAH composition exhibited striking similarities between indoor and outdoor environments, with medium molecular weight (MMW) and high molecular weight (HMW) PAHs dominating the profile. The I/O ratios of PAHs were less than 1, indicating a significant infiltration of these compounds from the outdoor environment into the indoor space. Correlation analysis showed that temperature played a pivotal role in influencing both PAH concentrations and I/O ratios. Source apportionment analysis identified that vehicle emissions as the primary sources of outdoor PAHs, which subsequently impacted indoor PAH levels.

The health risk assessment was particularly concerning, as the TEQBaP values exceeded the European Commission’s standard of 1 ng/m3, reaching 2.78 ng/m3 indoors and 3.57 ng/m3 outdoors. Moreover, the ILCRTotal values surpassed the acceptable level of 10E-6, highlighting the potential health risks associated with PAH exposure. It is noteworthy that the values of TEQBaP, HI, and ILCRTotal estimated in this study are all likely underestimated due to the incomplete consideration of other seasons, particularly winter. Despite these potential underestimations, the study still underscores the potential adverse health effects associated with PAH exposure. Therefore, further research is imperative to provide a more comprehensive assessment by accounting for seasonal variations.

Supplemental Information

Supplemental Information 1 The data for Figures

Supplemental Information 2 Supplementary Figures and Tables

Additional Information and Declarations

Competing Interests

Author Contributions

Data Availability

Xinfeng Wang is an Academic Editor for PeerJ.

Xiaomei Gao conceived and designed the experiments, authored or reviewed drafts of the article, and approved the final draft.

Ziyi Wang performed the experiments, analyzed the data, prepared figures and/or tables, and approved the final draft.

Xiaoyan Sun performed the experiments, analyzed the data, prepared figures and/or tables, and approved the final draft.

Weidong Gao conceived and designed the experiments, authored or reviewed drafts of the article, and approved the final draft.

Wei Jiang analyzed the data, prepared figures and/or tables, and approved the final draft.

Xi Wang performed the experiments, prepared figures and/or tables, and approved the final draft.

Fenfen Zhang performed the experiments, prepared figures and/or tables, and approved the final draft.

Xinfeng Wang conceived and designed the experiments, authored or reviewed drafts of the article, and approved the final draft.

Lingxiao Yang conceived and designed the experiments, authored or reviewed drafts of the article, and approved the final draft.

Yang Zhou analyzed the data, authored or reviewed drafts of the article, and approved the final draft.

The following information was supplied regarding data availability:

The raw data are available in the Supplementary File.

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
