# Peer review of "Characteristics, source apportionment and health risks of indoor and outdoor fine particle-bound polycyclic aromatic hydrocarbons in Jinan, North China"

_PeerJ, doi:10.7717/peerj.18553_

## Round 0.1 · original submission · Major Revisions

We have received three reviewers' comments for your manuscript, please see below and respond to them in detail

Reviewer 1 ·

Basic reporting

The paper discusses the ambient air concentrations of PAHs outside and outside Jinan University (China R.P.), and the associated cancer and non-cancer risk for humans. Intrinsic limitations of the study (overall, the sampling campaigns conducted in two only seasons) have been mentioned in the text.
Literature cited is sufficient, and sufficiently new.
All tables and figures seem necessary.

Experimental design

The study and the results presented fit with the Aims and Scope of PeerJ.
The investigation is well conducted and described.
The overall methodology applied in the study looks fine and sufficiently presented. It can be replicated elsewhere. Ethical standards seem to be met.
The results are of concern.

Validity of the findings

The main limitation of the study is that in-field campaigns were conducted only in two seasons. The experimental package was sufficient to demonstrate that people were exposed to both carcinogenic and non-carcinogenic risk at significant level, and by consequence to address policy aimed at preserving health anf mitigating pollution. On the other hand, the lack of measurements in winter and spring prevented a more precise quantification of risks. Hopefully, the research in this direction will continue.
As for data collected, they look robust, statistically sound and controlled.
Conclusions seem well stated and linked to purpose of the study

Additional comments

General Comments:
The paper discusses the ambient air concentrations of PAHs outside and outside Jinan University (China R.P.), and the associated cancer and non-cancer risk for humans. Intrinsic limitations of the study (overall, the sampling campaigns conducted in two only seasons) have been mentioned in the text.
The paper is well conceived and the work conducted is finely presented; besides, the results of this study can be helpful in implementing of better finalizing actions aimed at mitigating PAH pollution in the study area. All editorial criteria established by the PeerJ Editorial Board seem to be met.
By consequence, the paper merits consideration to publish in PeerJ.
Nevertheless, some minor points seem be taken in account before accepting the manuscript, which seems to require minor improvements in order to publish.
They are:
1. English language seems requiring some improvements. For instance, everywhere “PAHs” is used before “content”, “load” “concentration”, “ratio”… it must be replaced by “PAH”. Besides, in some cases the article used could be cut (e.g., using “PAHs” instead of “PAHs”, and “PCA” instead of “The PCA”; that makes the phrases more fluent.
2. All numbers reported (concentrations, percentages, etc.), would not hold more than three significant figures. Due to overall uncertainty of all measurements conducted in environmental contexts, more significant figures are poor of meaning.
3. Figure numbers should homogenized in the text and in the respective files (Figures 1-6 or 1A-1B, 2A-2B and 3A-3B).

Annotated reviews are not available for download in order to protect the identity of reviewers who chose to remain anonymous.

Reviewer 2 ·

Basic reporting

This study investigated particulate PAHs in indoor and outdoor air from a vacant office in Jinan, China. The measurement campaign was conducted in both summer and autumn of 2019 to evaluate seasonal differences in the pollution levels and sources. While the study did add some useful results in promoting indoor and outdoor relationship of particulate PAHs, the manuscript in its present form should be clarified and revised before its publication. Some of my main concerns are as follow:

Experimental design

Quality controls in PAHs analysis should be provided.

Validity of the findings

The significance and novelty of the study should be clear. PAHs, in either gaseous, particulate phases, or both, have been widely studied in numerous studies for many years, including but not limited to the levels, sources, and health risks in air, soil, water, etc., The contribution of this study, from a vacant office, should be clarified and clearer in its scientific implication and novelty. Besides outdoor PAHs, indoor air was also studied for many years, in either urban or rural homes (lines 70-73).

It is often believed that the pollution between summer and winter has distinct characteristics, and thus in many studies on seasonal differences, if not all four seasons, winter and summer samples were collected. Why the present study collected samples in autumn to analyze the seasonal differences?

Giving seasonal or temporal differences, the exposure level, which is important in the calculation of health risk, should take this into the consideration. And if only summer and autumn levels were available, there were significant biases, likely underestimates, in the health risk results.

The manuscript has a discussion on the abovementioned limitations, but insufficient and can hardly convince readers to accept the study significance. For instance, what’s the implication of a study on an office without human activities? How would be the results different in winter from that in autumn?

Lines 74-78, this may not for those with strong indoor combustion sources.



One notable limitation is that only one vacant office. What’s the representativeness of this site?

Additional comments

Lines 182-190, were these studies also only particulate PAHs, and analyzed 19 compounds? Would be the difference in sampling and laboratory analysis method influence the comparison results and conclusion?

Line 233, if p<0.05, was that considered “no significant difference”?


Do you have indoor or outdoor BaP standards for China? Besides the EU standard, WHO set guideline for BaP in indoor and outdoor air.

Most references in this paper, especially those in the introduction section, are old. Please update recent studies and refers.

Reviewer 3 ·

Basic reporting

Please see the attached file for the major comments

Experimental design

Please see the attached file for the major comments

Validity of the findings

Please see the attached file for the major comments

Additional comments

Please see the attached file for the major comments

Annotated reviews are not available for download in order to protect the identity of reviewers who chose to remain anonymous.

---

## Round 0.2 · accepted · Accept

The paper can be accepted according to the reviewers suggestions

Reviewer 2 ·

Basic reporting

the authors did a good job in revising this paper

Experimental design

well done with additional information added

Validity of the findings

results are clear and the relationship between indoor and outdoor air pollution is well interpreted

Additional comments

no

Reviewer 3 ·

Basic reporting

The authors have successfully addressed all the given comments earlier. Congratulations

Experimental design

The authors have successfully addressed all the given comments earlier. Congratulations

Validity of the findings

The authors have successfully addressed all the given comments earlier. Congratulations

Additional comments

Congratulations